

# On the optimal level of complexity for the representation of wetland systems in land surface models

Mennatullah T. Elrashidy[1], Andrew M. Ireson[1,2], Saman Razavi[1,2]

[1]Department of Civil, Environmental and Geological Engineering, University of Saskatchewan, Saskatoon, Canada
[2]School of Environment and Sustainability, University of Saskatchewan, Saskatoon, Canada

*Correspondence to*: Mennatullah T. Elrashidy (menna.elrashidy@usask.ca)



**Abstract.** Wetland systems are among the largest stores of carbon on the planet, most biologically diverse of all ecosystems,
and dominant controls of the hydrologic cycle. However, their representation in land surface models (LSMs), which are the
terrestrial lower boundary of Earth system models (ESMs) that inform climate actions, is limited. Here, we explore different
possible parametrizations to represent wetland-groundwater-upland interactions with varying levels of system and
computational complexity. We perform a series of numerical experiments that are informed by field observations from
wetlands in the well-instrumented White Gull Creek in Saskatchewan, in the boreal region of North America. We show that
the typical representation of wetlands in LSMs, which ignores interactions with groundwater and uplands, can be inadequate.
We show that the optimal level of model complexity depends on the land cover, soil type, and the ultimate modelling
purpose, being nowcasting and prediction, scenario analysis, or diagnostic learning.

## 1 Introduction

The Canadian boreal region covers about half of the land area of Canada. About 85% of all Canadian wetlands (~ 3 million
$Km^2$) are located in the boreal region (Mitsch, 1991). Wetlands are vital elements in landscapes as they can reduce the effect
of floods, store carbon from the atmosphere, improve water quality, absorb pollutants, and are considered the home of a wide
range of endangered wildlife and plants (Mitsch et al., 2013). Types of wetlands are bogs, fens, swamps, marshes, and
shallow water. Each type of wetlands differs from the others in terms of hydrology, water level, morphology, vegetation, and
biological aspects (Canada Committee on Ecological (Biophysical) Land Classification, 1988). Fens are wetlands that have
accumulated peat of over 40 cm, hydrologically interact with the surrounding groundwater and surface water, and have a
water level that is at or above the ground level for most of the year (Gingras et al., 2018). Fens constitute around 65% of the
peatland area within the boreal plain ecozone (A. R. Goodbrand, 2013). Fens critically depend on groundwater discharge
fluxes to sustain their moisture and water levels. Understanding the lateral hydrological interactions between groundwater
(GW) and surface water (SW) in wetland/fen systems is crucial to improve their representation in Land Surface Models
(LSMs) (Rivera, 2014). Such improvements can directly improve the simulation of land's energy and water balance as well
as different hydrological cycle components such as evaporation and streamflow (Blyth et al., 2021).
LSMs were originally proposed to estimate the vertical fluxes (energy and water) of the land surface, which is a necessary
lower boundary condition for climate models (Manabe, 1969). Over the past decades, these models have extensively been
modified to represent different processes such as soil moisture and vegetation dynamics. However, many recent studies have
highlighted the deficiencies in the current LSMs and discussed the scientific motivation to improve their process
representations (Clark et al., 2015; Davison et al., 2016; Fan et al., 2019). Lateral water movement, groundwater dynamics,
wetland hydrology, hillslope hydrology, and GW-SW interactions are examples of the elements which are either missing or
need more realistic representation. Typically, the coupling between different processes can be represented by three
approaches, which are uncoupled, one-way coupled, or two-way coupled. The choice of the suitable approach depends on
the required complexity level that can predict the variables of interest (Ogden, 2021).



Various studies focused on coupling (two-way interaction) between LSMs and GW models to investigate the effect of
groundwater dynamics on the simulated water and energy balance components (Kollet & Maxwell, 2008; Maxwell & Miller,
2005; Sridhar et al., 2018; Zhang et al., 2020). For example, Maxwell & Miller (2005) coupled the Common Land Model
(Dai et al., 2003) and the PARFLOW GW model. Their coupled model had a better simulation of soil moisture, runoff, and
GW dynamics than the uncoupled Common Land Model. The coupled model showed high dependency between
groundwater dynamics and mass and energy balance (Kollet & Maxwell, 2008). However, the difference in ET simulations
of their coupled and uncoupled models was negligible. In another example, Sridhar et al. (2018) coupled Variable Infiltration
Capacity (VIC) and MODFLOW models. Their coupled model showed significant improvement in simulated streamflow,
while the uncoupled VIC model was not able to predict the seasonal spring discharges in the streamflow. LSMs models were
also coupled with climate models, thereby improving most of simulated climate variables such as precipitation, relative
humidity, and surface temperature (Larsen et al., 2014; Maxwell et al., 2007).
Generally, the literature shows that the coupling of different models can improve the simulation of various hydrological
cycle components such as runoff, soil moisture, and water table fluctuations. However, such 'full' coupling approaches can
be quite complex both systematically and computationally, often rendering their applications impractical. In addition,
constraints in data availability, particularly around sub-surface processes, can limit the applicability of complex approaches.
Therefore, in practice, more parsimonious approaches to accounting the SW-GW interactions may be 'optimal' for a
modelling purpose of interest (Blyth et al., 2021; Ogden, 2021). What has remained elusive, however, is a thorough
characterization of tradeoffs between model complexity and adequacy to represent SW-GW processes for a particular
landscape and variables of interest (Yalew et al., 2018). Another factor that affects the selection of optimal model
complexity is the ultimate modelling purpose. Models can be used for: 1) nowcasting and prediction, which focuses on
simulating and predicting the expected behaviour of the system of interest in the near future, 2) scenario analysis, wherein
the model simulates the system under long-term changing conditions, and 3) diagnostic learning, which focuses on
understanding how the system behaved during a historic period (Razavi et al., 2022). Ultimately, the selection of optimal
model complexity will depend on the specific modelling task, the available data, and the desired level of accuracy.
Here, to address this gap, we aim to characterize the optimal level of complexity to represent the interactions between
uplands, groundwater, and wetlands in different landscape configurations. To this end, we run a series of modelling
experiments through four approaches, ranging from a full disconnect to full coupling. We particularly focus on a well-
instrumented and studied fen system in the Boreal region of North America. Based on lessons learned from these
experiments, we provide some recommendations on how to improve LSMs to better represent the important processes
around wetland systems – processes that are largely missing or poorly represented in the current generation of LSMs.

## 2 Study Area and Data

The study area is located within the White Gull Creek basin (WGCB), located north of Prince Albert, Saskatchewan (Barr et al., 2012), shown in Figure 1. The study area and transect are set using the Canadian Digital Elevation Model (CDEM, (Canadian Digital Elevation Model, 1945-2011 - Open Government Portal, 2022)).

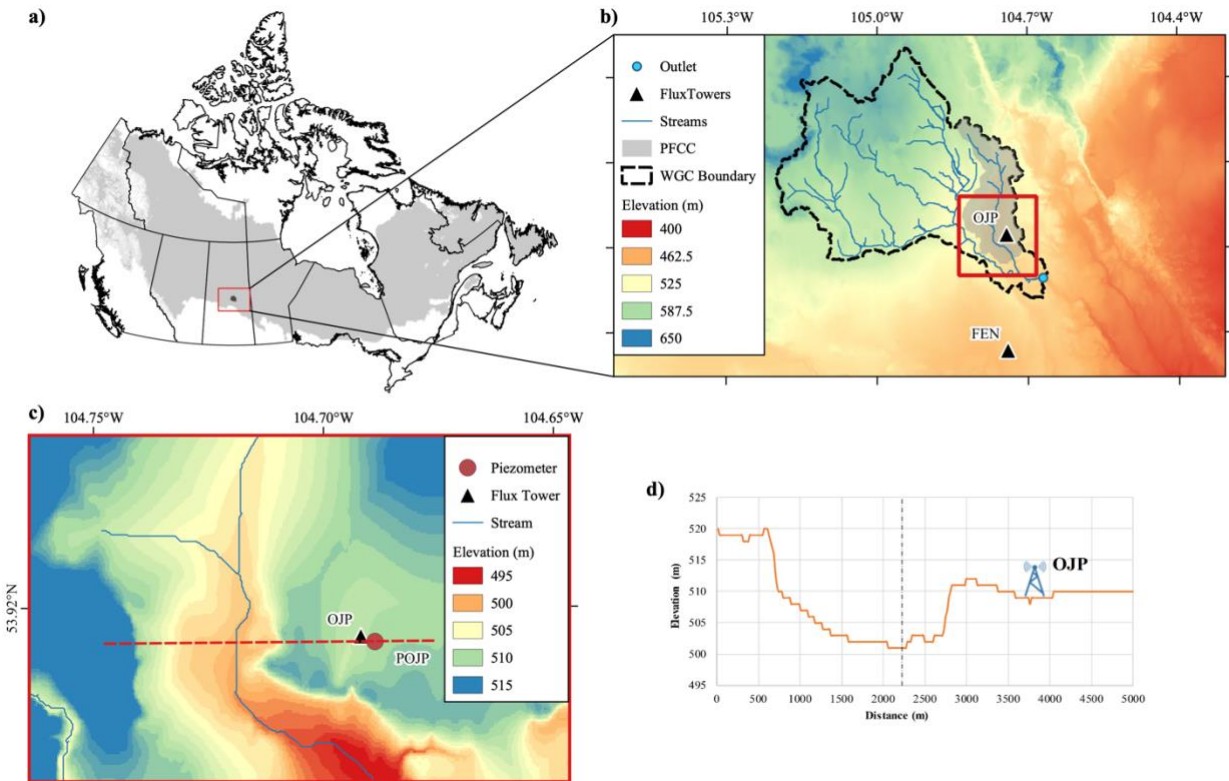

**Figure 1: Detailed view of the study area. a) Canada's boreal region in a gray shade, b) Focused view on White Gull Creek basin (WGCB) area and the two flux towers used in the study (Old Jack Pine (OJP) and Fen). The shaded gray area represents the Pine Fen Creek Catchment (PFCC) that include the Pine Fen (PF), c) Focused view on OJP area, the Piezometer of OJP (POJP), and the Pine Fen (PF), d) Cross-section of OJP and the PF (the dashed black line is assumed to be the line of symmetry).**

The upland is the area around the Old Jack Pine (OJP) flux tower (Latitude 53.92° N and Longitude 104.69° W) (Figure 1), which is part of the Boreal Ecosystem Research and Monitoring Sites (BERMS). The upland transect ends at the boundaries of WGCB, and the OJP site is located roughly in the middle of the transect (Figure 1-d), and is collocated with the piezometer (POJP, Figure 1-c) that is used to calibrate and validate the model performance. At the OJP site, the dominant land cover is Jack pine (Pinus banksiana Lamb) and the soil is sandy textured, which has poor nutrition and high drainage with a water table around 5m below ground (Barr et al., 2012). The meteorological data at OJP are available each 30 min for



23 years from 1997 to 2018. The groundwater table (GWT) observations are available at POJP location (Figure 1) from 2003
to 2018.
The lowland part of our transect is a fen known as Pine Fen (PF) which is located in Pine Fen Creek catchment (PFCC), a
tributary of WGC (39.9% of PFCC land cover is peatland) (A. Goodbrand et al., 2019). The PF site is a peatland mosaic
surrounded by forests of jack pine and black spruce. The average peat thickness is 0.65m with a maximum depth of about
2m. Unfortunately, we do not have direct meteorological observations from PF, and therefore as a proxy for this we use
observations from the BERMS fen flux tower site (FEN hereafter). FEN is located just outside WGCB boundary, about 8 km
south of the basin (Latitude 53.78° N and Longitude 104.69° W) (Figure 1-b). The FEN and PF sites are similar in terms of
the peat-soils, vegetation, and topography, so the FEN is considered a reasonable proxy for the vertical fluxes at the PF. At
the FEN, forcing data are recorded with a 30-minute resolution from 2003 to 2018, and the observed evapotranspiration rates
(ET) are from 2004 to 2010 and from 2013 to 2019 at 30 min intervals.

## 96    3 Upland-Groundwater-Fen Model Description

### 97    3.1 Conceptual Background

Our study is based on a real field site, using real field observations, as described in the previous section. However, this site is
not a perfectly constrained hillslope-fen system, i.e., a hillslope with 1D horizontal flow and a no-flow boundary condition at
the interfluve. Therefore, we use an abstracted hypothetical hillslope configuration to simulate the vertical and lateral flows
of water between atmosphere-upland-groundwater-fen system. This configuration is physically realistic, allowing us to test
the implications of different hillslope-fen coupling mechanisms in a controlled manner Figure 2.
Our model has three distinct components: (1) the upland soil water balance, which generates groundwater recharge and
runoff; (2) a lateral groundwater flow model beneath the upland, that may discharge water into the fen; and (3) a simple fen
water balance model, that receives inputs from rainfall, snowmelt, runoff and lateral groundwater discharge (in some cases),
and loses water to evaporation and discharge into a stream channel.
The model is driven by site observations of precipitation and other meteorological variables. A typical LSM is used to
simulate evapotranspiration fluxes in the upland and fen, groundwater recharge, runoff fluxes from the upland, and snowmelt
into the fen. The water table and groundwater discharge are simulated by a simple 1D unconfined aquifer model. The water
level at the fen and its discharge flux into an adjacent stream are simulated by a simple fen water balance model. The
connections between the three models of upland, groundwater and fen are configured in four ways as shown in Figure 2,
resulting in four different collective models as follows:
• V0) Uncoupled upland-fen model: The upland soil water balance and the fen are simulated as independent of one

114        another, and there is no groundwater model. Discharge from the upland in the form of surface runoff and soil





drainage (baseflow) are combined with the fen discharge and routed into the river. This configuration is
representative of many LSMs.
• V1) Uncoupled upland-groundwater-fen model: Soil drainage from the upland recharges the unconfined aquifer.
Surface runoff from the upland, groundwater discharge and fen discharge are combined and routed into the river.
The upland and the fen are again completely independent of one another. This configuration is representative of an
LSM that has a groundwater store with one-way vertical connection with soil column such as Community Land
Model (CLM) and Variable Infiltration Capacity (VIC) models (Clark et al., 2015).
• V2) Chained model: Soil drainage from the upland recharges the unconfined aquifer and groundwater discharge
contributes to storage into the fen. Surface runoff from the upland also goes into the fen, but this is typically a much
smaller flux compared to the groundwater discharge. Discharge from the fen is routed into the river. The
groundwater is independent of the fen, but the fen depends on discharge from the groundwater.
• V3) Coupled model: Soil drainage from the upland recharges the unconfined aquifer. Groundwater discharge into
the fen is determined based on head gradient between the groundwater and fen, and two-way water exchange is
considered between the groundwater and fen. Surface runoff from the upland also goes into the fen. Discharge from
the fen is routed into the river. The groundwater and fen are mutually dependent on one another.

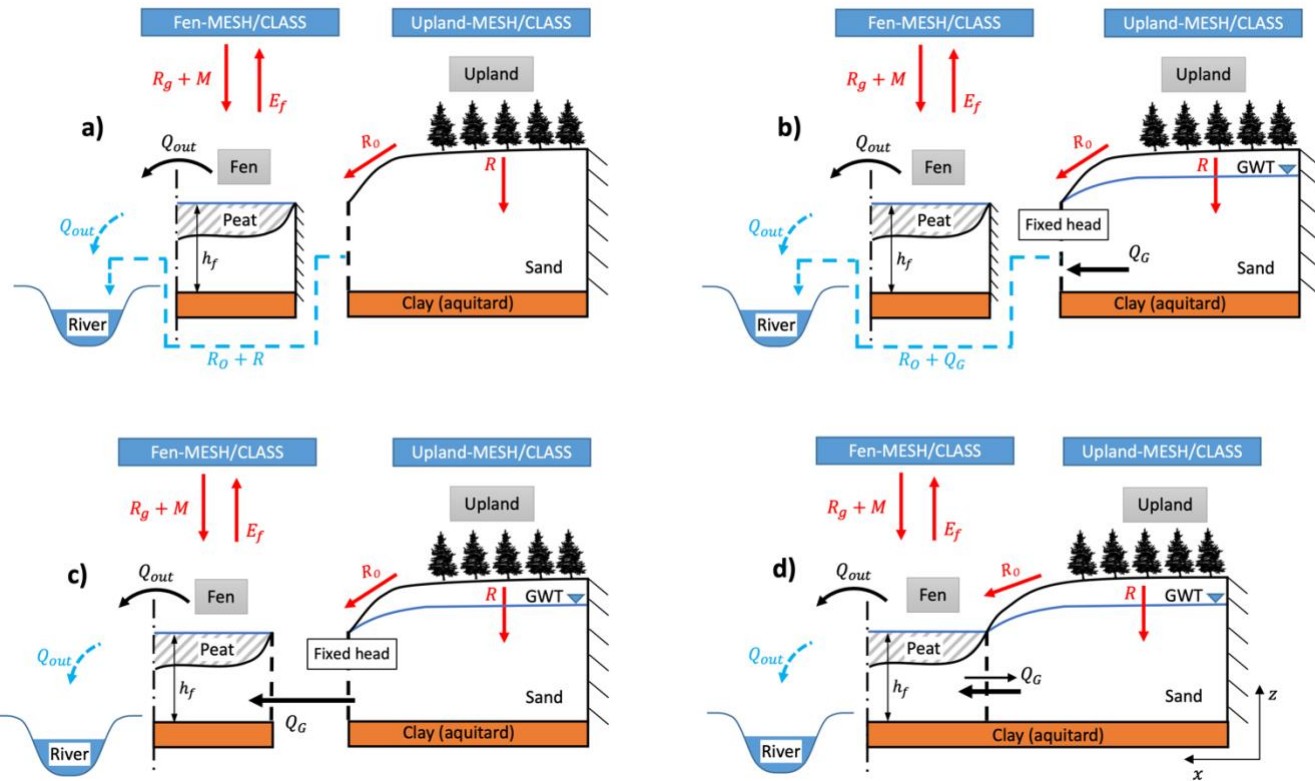

**Figure 2: Schematic of the different scenarios of representing the connection between the upland, GW, and fen. a) Uncoupled upland-fen and no GW (V0), b) Uncoupled upland-GW-fen (V1), c) Chained model (V2), d) Coupled model (V3). The vertical fluxes are rain on the ground ($R_g$), snowmelt ($M$), Evapotranspiration from the fen ($E_f$), and soil drainage as recharge into groundwater ($R$). The lateral fluxes are upland runoff ($R_O$), and groundwater discharge ($Q_G$).**

### 3.2 Upland Soil Water Balance Model

The upland soil water balance is simulated using the MESH-CLASS land surface model (Canadian Land-surface scheme; (Pietroniro et al., 2007; Wheater et al., 2022)) in a point-scale setup. The land cover of the grid cell is represented by one canopy type which is evergreen needle leaf, as most of the vegetation in the OJP area is jack pine trees. The model is forced using seven metrological components (precipitation, shortwave radiation, long-wave radiation, wind speed, specific humidity, temperature, and atmospheric pressure), which are collected from the OJP flux tower from 1997 to 2018 every 30 minutes. The soil depth is assumed to equal 4.1 m and is divided into three layers (i.e., the default CLASS configuration). We are mainly interested in two outputs from the model: Soil drainage ($R$) is used as either baseflow to the river (V0) or recharge to the groundwater (V1-V3); and surface runoff ($R_o$) is used as surface water input to river (V0, V1) or input to the fen (V2, V3).





### 3.3 Upland-Groundwater Model

The groundwater underneath the upland zone is represented as a 1-D horizontal unconfined aquifer, bound by a no-flow boundary on the right (the groundwater divide) and the fen on the left (Figure 1). The problem is governed by 1-D Boussinesq equation:

$$S_y \frac{\partial h}{\partial t} = \frac{K}{2} \frac{\partial^2 h^2}{\partial x^2} + R \,, \tag{1}$$

Where, $S_y$ (unitless) is the specific yield, $h$ (m) is the head in the aquifer (water table level), $t$ (days) is time, $K$ (m/day) is the lateral hydraulic conductivity, and $R$ (m/day) is the recharge rate.

Equation 1 is solved numerically using a block-cantered finite difference solution, integrated in time using the method of lines with the SciPy ode solver "odeint" (Virtanen et al., 2020), and the solution is coded up in python. The lateral fluxes are calculated at cell boundaries using Darcy's Law. The initial condition is assumed to be a uniform hydraulic head in the aquifer at the same water level as in the fen ($h_f$). The right-hand boundary condition is considered as a groundwater divide, and thus, a no-flow condition ($q(x = 0, 0 <= t <= t_{max}) = 0$). The flux on the left-hand boundary is determined using either a fixed head boundary (V1, V2) or based on the head gradient between the groundwater and the fen (V3). The groundwater model is driven by the recharge fluxes ($R$), which are output from the upland soil water balance model and are assumed to be spatially uniform.

### 3.4 Fen Water Balance Model

The fen is modelled as a simple lumped store, with the water balance equation.

$$n_f w_f \frac{\partial h_f}{\partial t} = (R_g + M - E_f)w_f + R_O \, L + Q_G + Q_{in} - Q_{out} \,, \tag{2}$$

Where, $n_f$ (unitless) is the porosity of the fen's material, $w_f$ (m) is the width of the fen, $h_f$ (m) is the fen's water level, $R_g, M, E_f$ (m/day) are the rainfall, snowmelt, and evaporation, respectively, $R_O$ (m³/day) is runoff from the upland, $Q_G$ (m³/day) is the lateral groundwater inflow, and $Q_{out}$ (m³/day) is the outflow from the fen. $Q_{out}$ is generated when a storage threshold, $h_{spill}$ is exceeded, and is assumed to be a non-linear function of storage above $h_{spill}$ given by:

$$Q_{out} = w_f \, c_{spill}(h_{spill} - h_f)^n \,, \tag{3}$$

Where, $c_{spill}$ and $n$ are parameters that control the outflow.

For $E_f$, $R_G$, and $M$, we assume that fluxes simulated by the MESH/CLASS model for the FEN site, located outside the watershed, are reasonably representative of the PF site. The forcing data (2003 to 2018) from the FEN flux tower are used to drive the MESH-FEN model. To simulate the peatland in the MESH/CLASS model, the land cover is assumed to be grass and the soil type is set to organic soil with three soil layers of changing properties as fibric, hemic, and sapric (Letts et al.,





2000). We manually calibrate the minimum stomatal resistance ($r_{s,min} = 100$ s/m), using local observations of the driving
meteorological variables. We compare the simulated $E_f$ with both the observed fluxes and with potential evaporation
calculated using the Penman Monteith equation, and we found all three are consistent, showing that the evaporation from the
FEN is unstressed (i.e., not water-limited). The values of $Q_G$ are either zero (V0, V1) or equal to the groundwater discharge
at the right-hand boundary (V2, V3).
**4 Model Analysis and Performance Evaluation**
**4.1 Calibration strategy**
The performance of the different collective models is evaluated using the GWT observations at the POJP location by
calculating the root mean squared error (RMSE)). For the upland water balance model, we use the same calibrated
parameters from the study of (Nazarbakhsh et al., 2020), in which they use the CLASS model to assess the controls of
evapotranspiration in the seasonally frozen forest. For the other parameters, Monte-Carlo simulations with 15,000
realizations (randomly generated parameters from the feasible parameter space in Table 1) are used to run the uncoupled
upland-GW (V1). The behavioural runs are identified as the realizations with RMSE <0.08m (threshold is chosen rather
arbitrary based on expert judgement and calculated for the period from 2003 to 2009) and are used to perform the uncertainty
analysis of GWT simulation. The parameter set with lowest RMSE is considered the calibrated parameter set and is used to
validate the model. This parameter set is also used to run all the other collective models throughout the study.
**Table 1: Monte-Carlo analysis parameters ranges of the uncoupled upland model.**

| Parameters | Description | Lower bound | Upper bound |
|---|---|---|---|
| $log(K)(m/d)$ | Hydraulic conductivity Logarithm | -1 | 3 |
| $S_y$ | Specific yield | 0.1 | 0.5 |
| $h_f(m)$ | Fen's water head | 5 | 20 |
| $L(m)$ | Hillslope length | 3000 | 3500 |
| $x(m)$ | Piezometer location | 1700 | 1800 |





## 4.2 Effect of different upland properties

We develop two hypothetical numerical experiments to explore the conditions under which different levels of model complexity may be necessary. Experiment 1 focuses on hillslope geometry, by considering five different values of the hillslope length ($L$, that is the length of the upland). $L$ values of 100, 300, 500, 1000, 2000, and 3000 m are considered, and all other parameters are the same as in the original model setup. This experiment uses the chained (V2) and the coupled (V3) versions of the model. Experiment 2 focuses on soil properties, by comparing the original (sandy soil) setup with a fine-grained soil representative of mineral soil which typically can be found in the prairies area. This is achieved using an alternative configuration of the MESH model, in which the parameters are changed with values to represent a grassland cover and a fine-grained soil texture (clayey soil) resulting in different amounts of runoff, infiltration and soil drainage. In the upland algorithm, the values of $K$ and $S_y$ are set to 1.35m/d and 0.1, respectively, to characterize the fine-grained soil. The simulated GWT and GW fluxes using the new MESH/CLASS run (representing fine-grained soil) are compared with the model results when using the original study setup (upland algorithm forced with MESH-OJP).

## 5 Results and Discussion

We assess the performance of the model in the upland (Section 5.1) and fen (Section 5.2) independently. Next in Section 5.3, we assess the sensitivity of the simulated outflow from the integrated upland-groundwater-fen system, which corresponds to the outflow on a grid cell scale in LSMs (i.e., the bulk system outflow) to the different model configurations. Lastly, in Section 5.4 we describe the results of the two numerical experiments exploring upland properties.

### 5.1 Model performance in the upland

In the upland, the performance of the vertical land surface fluxes is explored by (Nazarbakhsh et al., 2020). We are able to assess the upland model's performance in reproducing observations of the water table elevation. As explained earlier, there is no groundwater in V0 of the model. In the uncoupled (V1) and chained (V2) models, the groundwater simulations are identical. In the coupled model (V3) the water table simulations may, in principle, differ from those in V1 and V2. Therefore, here we compare V1 and V3 separately.

### 5.1.1 Uncoupled model calibration and validation

The upland component in the uncoupled model is driven by the recharge values that are generated using the MESH-OJP model (Figure 3-a). Figure 3-b shows a comparison between the simulated and observed GWT at OJP site using the uncoupled upland model. The behavioural runs (69 runs with RMSE<0.08m) from Monte-Carlo analysis are used to generate the uncertainty bounds in the simulation of both GW discharge and GWT (Figure 3). It can be seen from Figure 3-a that the



recharge and the simulated GW discharge responded to each other consistently. Also, the changes in the GWT corresponded
to both the recharge and the GW discharge peaks.

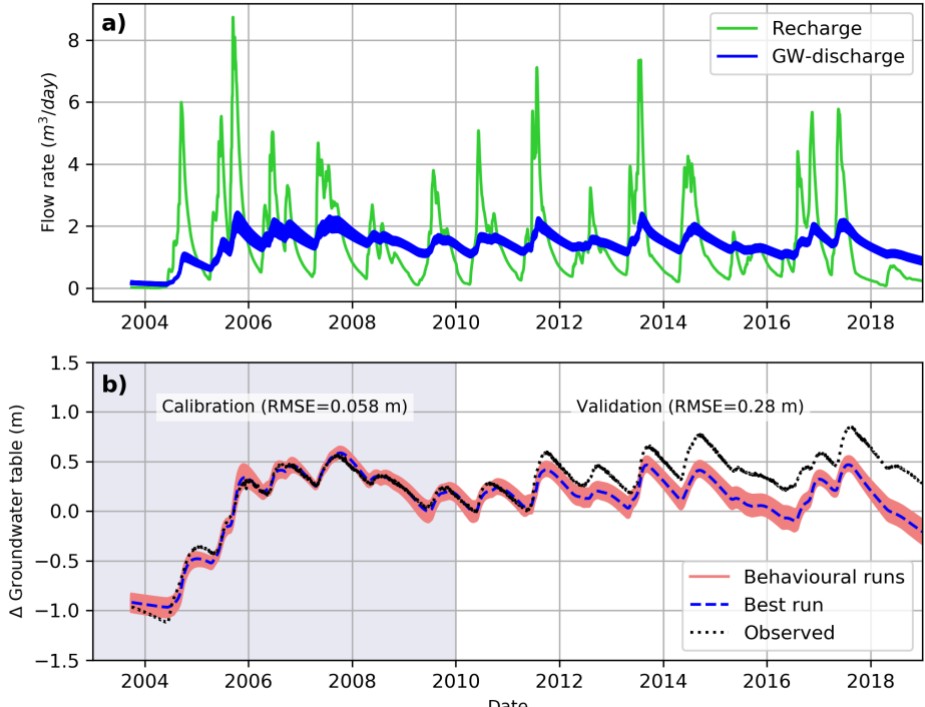


**Figure 3: a) Upland recharge rates into the groundwater aquifer that are generated using MESH-OJP model and are used to drive**
**the upland component, b) comparison between the simulated and observed changes in the GWT at OJP site. The uncertainty**
**results (GW discharge and GWT) are obtained from the behavioural realizations using Monte-Carlo analysis.**
For the uncoupled upland model, the best value of RMSE is equal to 0.058 m for calibration, and the value for the validation
is 0.28 m (corresponds to the parameters values as $K = 135.1$ m/d, $S_y = 0.24$, $h_f = 9$m, L = 3275m, and $x = 1733$). The
uncoupled upland model is able to simulate the GWT in the calibration period with a narrow uncertainty bound. In the
validation period, the simulated GWT matched the observations until the spring of 2011, when a discrepancy is noticed, and
the GWT is underestimated thereafter. The GWT underestimation is caused by low recharge rates from 2011 to the end of
the simulation, which might be caused by either undercatch in the observed precipitation or problems with the
MESH/CLASS model in simulating the recharge rates at this period. The MESH/CLASS model problem could be because of
overestimation of evapotranspiration rates at the upland site, which means the MESH/CLASS model might need re-
calibration to a longer period of data. We should note that although the model showed underestimation of the GWT
magnitude (from 2011 to 2018), it captured the same pattern during the same period. We believe that our model performance
acceptably serves the purpose, as the main purpose of this study is to perform numerical experiments to help us understand
the upland-groundwater-fen system dynamics.





### 5.1.2 Upland-Uncoupled vs Coupled

Figure 4 shows the comparison between the simulated GWT using the uncoupled (V1) and coupled (V3) model versions and the observations. The overall performance of the coupled version showed only a slight improvement (RMSE=0.18m) over the uncoupled version (RMSE=0.22m). A comparison between the simulated uncoupled and coupled systems shows that, in the period of record, considering the effect of the fen system does not affect the simulated GWT underneath the upland in the case of OJP and PF. However, the impact of this coupling might become more profound for other sites with different settings or the same site under different climate conditions.

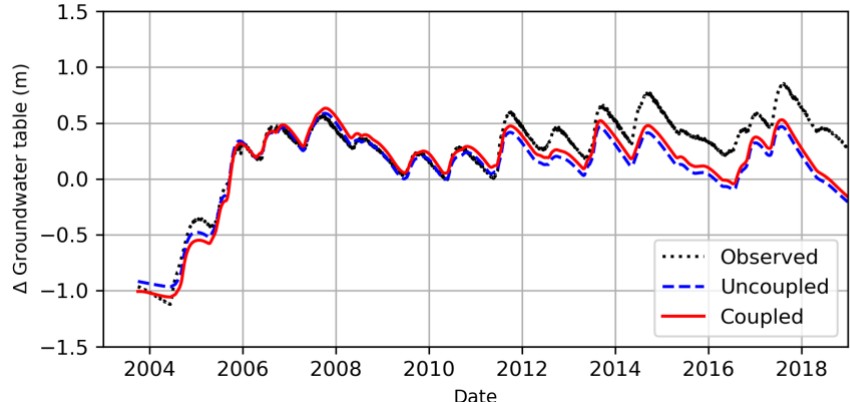

**Figure 4: Comparison between the simulated GWT using both the uncoupled and coupled models with the observations.**

### 5.2 Model performance in the fen

For the fen, we are not able to directly test our model performance due to a lack of data, and instead, we explore the sensitivity of the fluxes to the change in the modelling configuration (interaction between the upland and fen). The fen's outflow and changes in water level are compared for the three versions (V1, V2, and V3) in Figure 5. In the uncoupled (V1) model, when there is no groundwater inflow to the fen, the estimated outflow and water level changes are unrealistic as the outflow is almost zero and the water level kept decreasing from one year to another. On the other hand, the chained (V2) and coupled (V3) models had a reasonable simulation of the outflow and water level changes of the fen but with differences from each other particularly in terms of flow rate. The overall trends of flow rates of V2 and V3 look similar, but the higher-frequency features (e.g., daily flows) show different dynamics from time to time. Flow rates of V3 are affected by the two-way water exchange between the upland and fen, which are based on the variable fen water level, unlike the concept of constant fen water level that is used in V2. This is apparent in 2004, wherein V3 model generated negative flow rates (i.e., water flows from the fen into the upland). This comparison shows that the groundwater inflow from the upland into the fen cannot be ignored when simulating the fen, however, the chained modelling approach might be deemed adequate to capture the fen system dynamics. Coupled configuration is needed to study the short-term impacts and changes in the fen outflow.

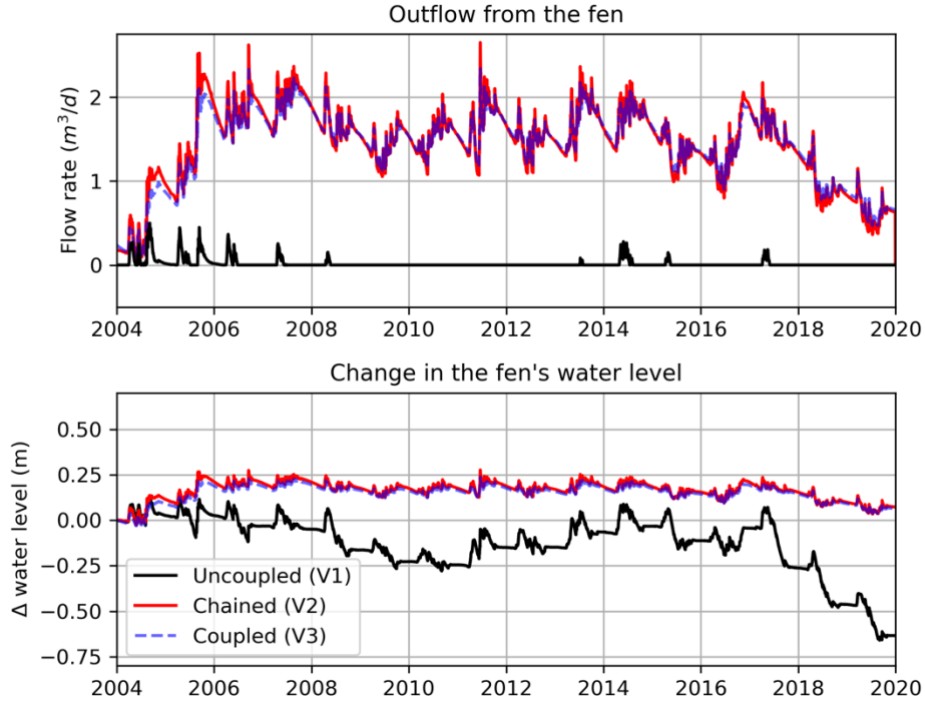

**Figure 5: Comparison between the fen's output (fen outflow and change in the fen water level) for the three modelling scenarios (uncoupled (V1), chained (V2), and coupled (V3)).**

**5.3 Outflow of The Integrated Upland-Fen System (Grid-cell scale)**

In this section, we investigated the total outflow from the integrated upland-GW-fen system as a whole unit under different modelling configurations, which are the possible approaches to represent such system in LSMs, to simulate the amount of the total flow that discharges into the river network (streamflow) (check blue dotted arrows in Figure 2). In LSMs, a grid cell can contain multiple components, such as upland and fen. In this case, the total outflow of the grid cell is the combined outflow from both the upland and fen components after considering the interaction between them based on the used modelling configuration. This is done to assess the optimal level of model complexity that can simulate the streamflow adequately.

In the chained (V2) and coupled (V3) models, the simulated GW discharge into the fen from both models are almost the same at a daily and annual scale (Figure 6). Also, the total outflow from the grid cell into river had no significant difference in the two cases. In the case of uncoupled upland-GW and fen model (V1), there is no GW discharge into the fen, but the daily and annual total outflow into the river is similar to that of V2 and V3. Therefore, on the grid cell scale, the interaction between upland and the fen had no effect on the total outflow from the grid cell and discharge into the river for our model configuration.





In case of V0, when there is no account for the GW storage and all the soil drainage (recharge) is considered as baseflow,
which is discharged directly into the river network (the case in most of the current LSMs). Therefore, the simulated outflow
(of V0) into the river is significantly different (with overall greater magnitudes) compared to the other three versions that
accounts for GW storage (Figure 6). That means considering the GW dynamics underneath the upland is essential.

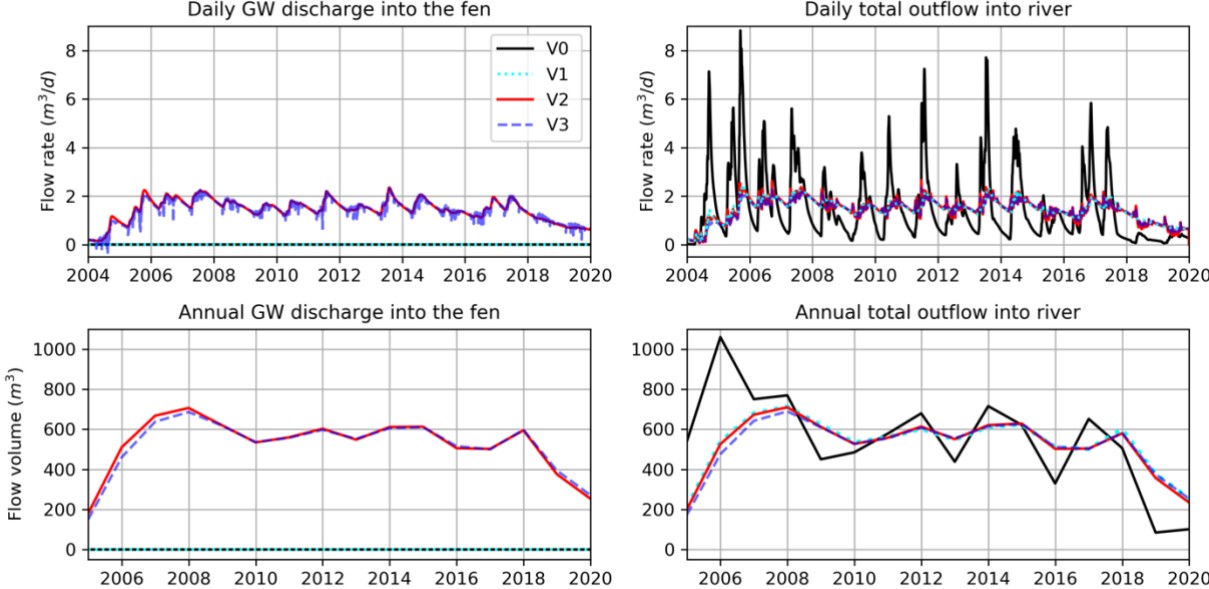


**Figure 6: Comparison between GW discharge into fen and total outflow form grid cell into river on daily and annual scales for V0,**
**V1, V2, and V3 modelling scenarios** (Error! Reference source not found.)

**5.4 The effect of different upland properties on the upland-fen interactions**

Here, we run two additional numerical experiments to explore the optimal level of modelling complexity for different upland
site properties. The experiments are hypothetical (no observations) and represent other possible sites' conditions.

**5.4.1 Experiment 1: Different hillslope lengths**

Figure 7 shows the simulated upland GWT using different upland hillslope lengths (width of the fen is constant) and
compares the results in the case of chained (V2) and coupled (V3) upland. In the case of horizontally large aquifers ($L >$
$1000m$), as in our original study setup, there is no significant difference in the simulated GWT and GW flux when using the
two model configurations. In contrast, in small hillslopes (lengths between 100 to 500 m), the chained model is not able to
reasonably capture the fluctuations of the upland GWT, as the simulated GWT is almost constant. Also, the difference can be
seen when comparing the simulated GW fluxes in the two model configurations (Figure 8). In small hillslopes, the coupled





model is able to capture the water amounts that move from the fen into the upland (negative flux values), which are
considerable amounts frequently present throughout the year.

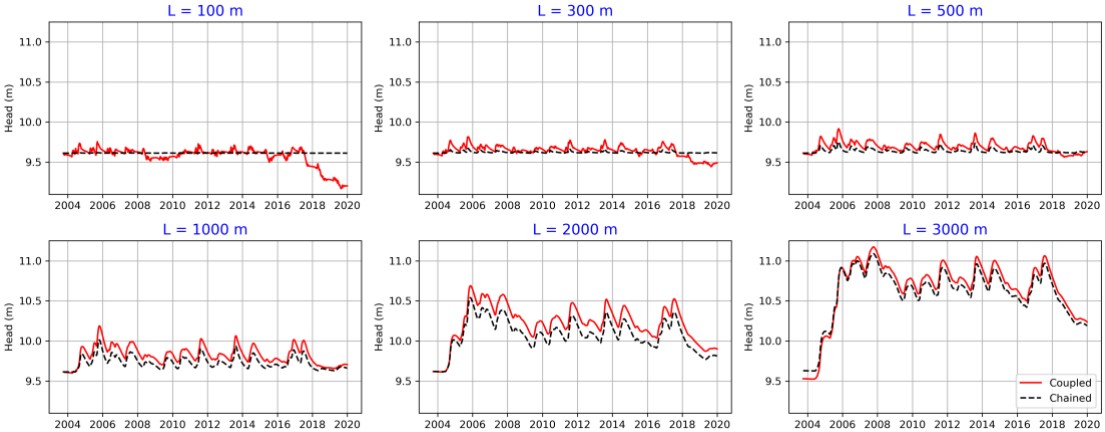


**Figure 7: Simulated upland GWT for both Chained and coupled model versions by using different hillslope length (*L*).**

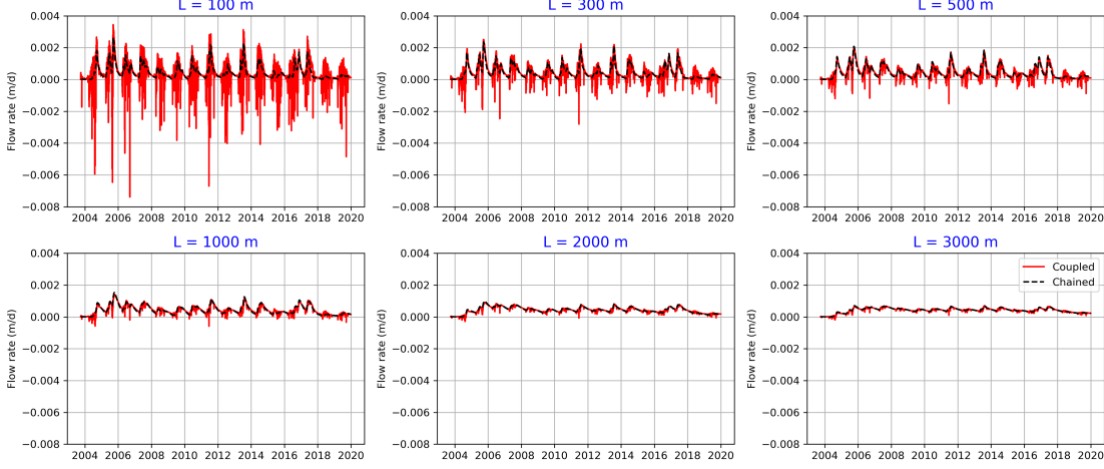


**Figure 8: Simulated upland GW fluxes for both chained and coupled model versions by using different hillslope length (*L*).**
In the case of large hillslopes, the groundwater size is significantly greater than the fen size (140m) and therefore large
amounts of water move from the upland into the fen. As a result, the upland controls the dynamics of the whole system. In
such cases, the upland can be simulated independently with no account for the interaction with the fen. In the case of small
hillslopes, however, the fen is the dominant contributor to the system as the water moves continually in two-way directions.
In such cases, the coupling between the upland and fen is essential.





### 5.4.2 Experiment 2: Different upland soil properties

Figure 9 shows a comparison between two upland cases, 1) coarse-grained soil (high permeability) and forest Land cover (high density vegetation) with a long hillslope length, which is our original study setup (Figure 9-a), and 2) fine-grained soil (low permeability) and grass land cover (Figure 9-b). In the case of coarse-grained soil (Figure 9-a), the main contribution to the upland GW system is the high recharge rates because of the high infiltrability of the soil (coarse-grained/sandy), and relatively very small amounts of surface runoff to the fen. Thus, in this case, the dominant component of the upland-fen system is the GW water fluxes from the upland into the fen (through the subsurface water movement). Water arrives as precipitation on the upland, infiltrates into the soil and recharges the aquifer, and finally moves laterally in the aquifer to discharge into the fen. To represent these system dynamics, the chained (V2) modelling approach for the upland system seems adequate to simulate the GW system (GWT and GW discharge) of the upland, as the difference between the results when using chained (V2) and coupled (V3) models is relatively small (Figure 9-a). However, the coupled (V3) model can simulate the dynamics of the daily GW flows due to the frequent change of the fen water level. Accordingly, the flow direction is reversed to be from the fen into the upland (negative flow values) from 2004 to 2005 (Figure 9-a).

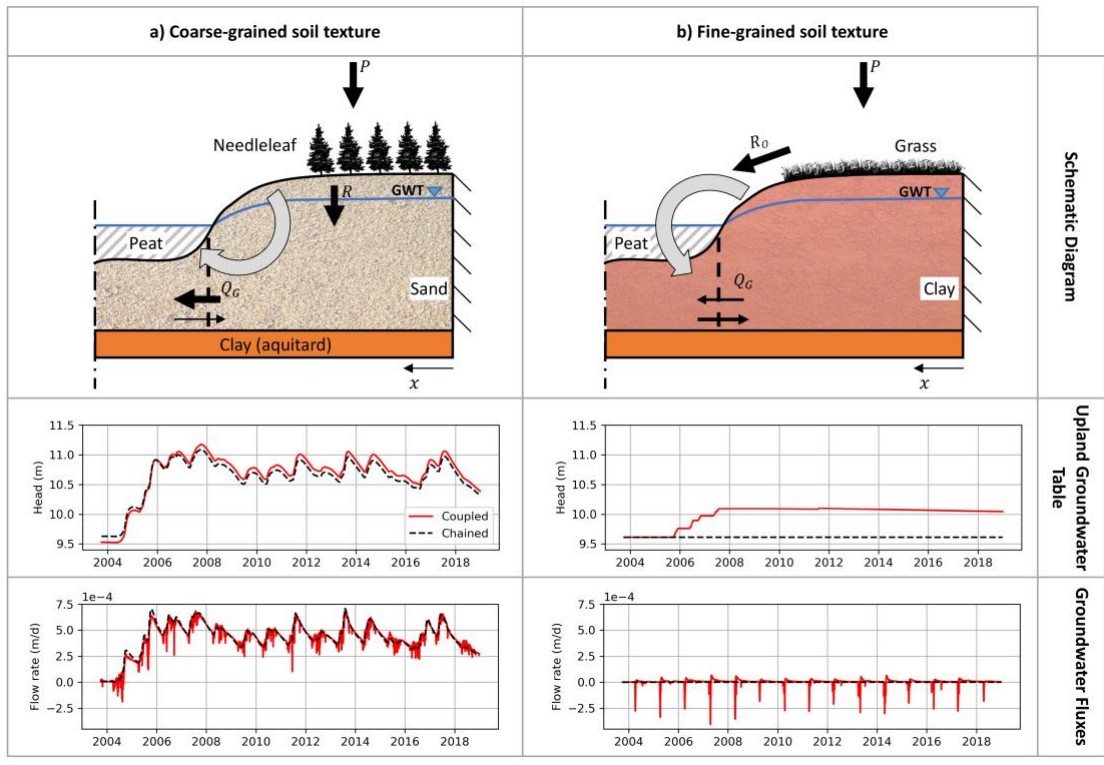

**Figure 9: Comparison between two different upland site conditions and simulated upland GWT at each case, a) OJP with coarse-grained soil texture and evergreen needleleaf canopy, b) St Denis with fine-grained soil texture and grass land cover.**





The systems dynamics are different when the soil has fine-grained texture, and the vegetation has low density (Figure 9-b). In this case, large amounts of surface runoff move directly from hillslope into the fen whereas very limited water may infiltrate into the upland aquifer. Figure 9-b shows that the chained model is not able to simulate any of the GW dynamics underneath the upland, whereas the coupled model captured the system's dynamics (Figure 9-b-GW Fluxes). The fen in this case is mainly fed by the surface runoff fluxes, then the water moves laterally into the upland (especially during snowmelt season). Hence, the main flow path is from the fen into the upland GW. That means, the upland GW dynamics are dominated by the subsurface water fluxes coming from the neighbouring wetlands/fens. It is obvious that considering only the chained approach (one-way exchange between upland and fen) in the case of fine-grained upland soil cannot reasonably capture the real dynamics of the system. However, the full coupling between the upland and the fen (two-way water exchange) allows the model to represent the actual dynamics of the upland aquifer underneath fine-grained soil layers.

## 6    Conclusions

The insights from applying alternative model configurations to the upland-fen system in this study are as follows:

1.  We are able to reasonably simulate the GW dynamics underneath the upland using the 1D Boussinesq equation. There are no significant differences between the coupled and uncoupled modelling approaches for simulating the upland water table elevation, because the dominant flow direction is from the upland to the fen.

2.  To simulate the water level in the fen, the GW input from the upland cannot be ignored. However, there is no significant difference between the chained (one-way interaction) and coupled (two-way interaction) approaches in terms of the simulated fen water level and outflow.

3.  The inclusion of upland-fen interactions had no significant impact on the discharge into the river network. However, inclusion of GW storage had a major impact on the timing and magnitude of river discharge.

We found that when the size of the fen is large relative to the upland, it is essential to use a coupled fen-upland modelling approach, as there can be substantial bi-directional exchanges of water between the fen and the upland GW at different times of the year. The coupled modelling approach is also more likely to be necessary when simulating uplands with fine-grained soils, as the fen receives more surface runoff and less groundwater input, and therefore, loses a significant amount of water into the groundwater system.

In general, if the main objective of the model is to simulate streamflow, coupling (two-way interaction) between the upland and fen/wetlands can likely be ignored. However, groundwater dynamics must be represented in LSMs as they significantly affect the total outflow (streamflow) from the whole system. On the other hand, if the simulation of the storage and fluxes within fen/wetlands are of interest, then the chained modelling approach is the least complex level needed to account for the contributions of the surrounding upland and GW systems.

This study gives insights into the necessary model complexity for simulating an upland-GW-fen system within land surface models. The outcomes of this study can help in improving process representation in LSMs and guide current and future



hydrological modelling practices in wetland/fen-dominated areas. This can lead to simulating the water cycle more accurately in that region, which would contribute to better water resources management and allocation and improve the LSMs' ability to predict the effect of future climate change on the wetlands.

**Data availability**

We used the Water Information Systems KISTERS (WISKI) data. The data is publicly available on https://wiki.usask.ca/display/GWFDM/WISKI and were downloaded using the python WISKI tool (https://github.com/incsanchezro/WISKI_Tools_GWF). The meteorological data are from Old Jack Pine (OJP) and Fen flux towers, and the upland groundwater observations from OJP.

**Author contribution**

**ME**: conceptualization, methodology, model development, writing; **AI**: conceptualization, methodology, model development and writing (review and edit); **SR**: conceptualization, methodology, writing (review and edit).

**Competing interests**

The authors declare that they have no conflict of interest.

**Acknowledgements**

We would like to thank the Global Water Futures (GWF) and the Integrated Modelling Program for Canada (IMPC) for funding this research.

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
