# Peer review of "On the optimal level of complexity for the representation of groundwater-dependent wetland systems in land surface models"

_Hydrology and Earth System Sciences, 2023_

## Author Comment (AC1)

**Response to Reviewer 1**

**General Comments**

Here the authors explore different model complexities and configurations to highlight the need for coupling upland-wetland interactions in land surface models to better capture downstream hydrologic fluxes. In general, the inclusion of wetlands in LSMs is an important consideration for many types of landscapes. Overall, the manuscript was clear and the approach was sound. Below are some points that I think the authors should consider to help clarify some points for the reader.

Dear Reviewer, we would like to thank you for your constructive comments on our manuscript. In this file we respond to each comment to address your concerns. The response for each point is written in blue.

One broad comment is that there is a bit of a disconnect in how the manuscript was framed (the title, abstract, introduction had a heavier emphasis on wetlands in general) vs the actual analysis and discussion (much larger emphasis on fens, which is the study site). The conclusions were logical based on the data from the fen, but it would be helpful to a) emphasize that fens by definition receive significant amounts of groundwater inputs and b) how other types of wetlands with different levels of GW connectivity could change your conclusions.

We agree that the framing of the manuscript can be improved. We focus mainly on groundwater dominated fen wetlands, since our case study is a fen, but we also conduct numerical experiments that look at non-groundwater dominated wetlands (section 5.4.2). We will edit the abstract, introduction and objectives to better explain this.

**Specific Comments**

- It would be helpful to reference more existing literature on the need for including proper interaction between wetlands and the upland to help strengthen the case for this particular study. While there may not be as many LSM studies that look at this directly, drawing parallels to watershed scale studies, which has quite a few studies in the recent years.

We agree that we can include more literature on watershed scale applications of LSS and will seek to do this in the revised manuscript.

- Minor suggestions for Figure 1
    - Expand part d to be larger to match the other panels for clarity;
    - add the location of POJP piezometer for clearer connection to panel c);
    - if possible, indicate the extent of the fen in part d) just to give some context for the reader. I understand that this can be variable throughout the fen. If the fen goes beyond the transect, ignore this comment!

We will include these points in the revised manuscript.

- L123: The statement regarding the amount of surface water vs groundwater into the fen seems something unique to this system, and isn't necessarily a feature of the V2 configuration –

consider putting this elsewhere. As a side note, because I don't have much context to the wetland:upland ratio or the water balance, the reader might be surprised by this statement. Might be worth indicating the relative areas for the wetland + upland and/or some estimated water balance in the site description

We agree that as the relative magnitude of the surface water flux is not a general feature of the V2 model configuration, so we will move this text from this part. Regarding the wetland-upland ratio, the total study domain is about 3300 m and the wetland width represent about 150 m of that length. We will focus on that point more in the study description section to make it clearer to the reader.

- L137: does the MESH-CLASS model have saturation-excess and infiltration-excess runoff components to determine runoff vs recharge? Similarly, how does it calculate the R flux? I would not expect a full description of the model, but since this is a major connection to the GW component, it would be helpful to briefly describe it. Also, is snow accumulation and melt modelled in the upland? I assume so, but since it's explicitly mentioned in the fen model, but not here, it can create confusion

The MESH/CLASS model uses an infiltration excess method (based on Green and Ampt) to calculate infiltration. Excess ponded water at the top of soil column is used to calculate overland runoff. We will mention this in the model description.

Regarding, snow accumulation and melt at the upland, they are simulated by the MESH model, and we will mention this in the revised manuscript. However, note that these fluxes are not used directly in our upland-wetland model – rather we use runoff and drainage generated from snowmelt by the MESH/CLASS model. For the fen the snowmelt fluxes are used directly in the fen water balance model.

- L148: I may be mistaken, but I don't think Figure 1 show the groundwater divide/no-flow boundary condition (this is in Fig 2?)

We assume the groundwater divide is the basin boundaries shown in Figure 1-b as black dashed line. We will further clarify that for more clearness in the revised manuscript.

- L151 and paragraph: would be helpful to mention how dx is determined within the Darcy's Law calculation

On Line 153 we added the following: "Equation 1 is solved numerically using a block-centered finite difference solution on a regularly spaced grid with dx = 0.01*L (L=Hillslope length).

- L165: Units are not consistent across equation. If Ro is upland runoff (m3/d) and L is the hillslope length (m), the units are [L4/T]; similarly the (Rg+M-Ef)wf component have units of [L2/T], while the Q terms are [L3/T].

Thanks for pointing out this mistake. The units of $RO$ should be m/d and for $Q_G$ and $Qout$ its m$^2$/d, knowing that we assume a unit area (1m) in y direction. That will be corrected in the revised manuscript.

- L168: Are Cspill, hspill, n calibrated parameters, and is the model sensitive to them? I believe they are not referenced again later on but one might assume that they have major roles in changing Q.

We agree we should provide the parameter values used, and explain how these values were selected, and will do this in the revised manuscript. These three parameters control the timing and magnitude of discharge from the fen into the river channel. We did not have streamflow data to calibrate these parameters, and therefore our approach was to use sensible fixed parameter values that enable us to perform controlled numerical experiments. Our objective is not to simulate streamflow precisely, but rather to explore the sensitive in simulations to changing coupling between the wetland and the upland.

h_spill represents the spilling threshold of the fen, so that when the fen water level is below h_spill, there is not discharge from the fen into the river. We set this value equal to the elevation of the fixed head boundary from the uncoupled groundwater model (Figure 2b/c).

The values of $c_{spill}$ and n ($c_{spill}$ = 0.1 and n = 1.5) were arbitrarily chosen within the reasonable ranges. The ranges were defined based on the recommended ranges in (Razavi and Gupta, 2019), which they used in modelling the fast reservoir with non-linear response in the HBV-SASK model.

We will make this clear in the revised manuscript.

- L181: Is there any downstream gage to calibrate? I would be hesitant to say that calibrating the GWT in the upland represents the performance of the collective fen-upland-GW models - especially since three of them do not have the backwards interaction in the model structure

We agree that calibrating the GWT is not sufficient to conclude that the model accurately represents our specific field site – however, the main objective in this study is to perform a comparison between the alternative modeling approaches with using the one reasonable parameter set (as we explained in the previous response).

- L185: were these all generated from uniform distributions?

Yes, the 15,000 realizations were generated using a uniform distribution. That will be added to the text.

- L186: maybe use "(threshold is chosen arbitrarily based on…)" instead of "chosen rather arbitrary"

This threshold value was selected based on identifiability analysis and considering the behavioural realizations, which is not included in the manuscript since it is out of scope.

- L189: I would consider putting in the best parameter set in Table 1 to give context for readers rather than just embedded in the text later on

Agreed - the table will be modified in the revised manuscript to include the calibrated parameter set.

- L194: are these L values corresponding to likely wetland-upland areas?

These values correspond to the length of the upland hillslope. At this case we explore the effect of changing the area of upland with having the same area for the fen.

- Figure 3: Should write in caption the simulation number (V1?)

Yes, these results are for model version V1. That will be added.

- Figure 6d: missing y axis label to be consistent

That should be the same label as in Figure 6c. However, that will be added for clarity.

- L291: It is slightly hard to understand why at low L values, the groundwater table does not fluctuate in the chained model – it would be worth discussing why. I would assume there's still stochastic inputs to the groundwater from recharge/precipitation, and it's not that the groundwater table has reached the lower boundary/bedrock

The water table does not fluctuate very much in the chained model when the hillslope length is small because of the fixed head boundary condition. In the coupled model, there is no fixed head boundary, so the water table fluctuates more.

- L306: Land is capitalized mid-sentence

That will be modified.

- L306: Because the reader does not know how the forest and grass (is it solely a runoff-coefficient difference, or does ET get affected too?) affects the model fluxes, it's hard to attribute the changes in the model to solely the soil properties, which is the focus of this section.

In our model, changing the vegetation impacts ET and runoff, while changing the soil properties changes the runoff vs infiltration, recharge, and ET. The first model configuration in this numerical experiment was our original setup (OJP in the southern boreal forest). The second configuration (grass over finer soil) was designed to maximize surface runoff, to provide a contrasting case with the groundwater dominated case. The configuration was based on a model configuration for a grassland site (St Denis in Ireson et al., 2022 - MESH point scale paper, in the prairies south of the forest). We did this because we have credible parameters for each configuration – they are not entirely hypothetical. We will explain this rationale more clearly in the manuscript and provide the citation.

- L313: I think that for this instance, it is true that the chained approach is adequate to illustrate the coarse grained soil texture. But I think it's worth commenting that in areas of smaller hillslopes/contributing areas, that may not be the case (as proven in your previous experiment)

We agree and will mention this at this point in the revised paper.

L346: I would not necessarily include wetlands in 'fen/wetlands' as fens by definition have a lot of GW inputs. Having wetlands here can cause readers to assume that wetlands that either have more bi-directional interactions with the upland via groundwater, or don't receive groundwater, should be treated the same way. While the authors wouldn't run more simulations to capture other wetland types, I think it's a valuable discussion point

We agree with you, this should be clearly discussed in the conclusions to prevent any confusion. We would say that this is an investigation and fens are taken as an example of wetlands that have more dependency on two-way groundwater exchange.

**References**

Ireson, A. M., Sanchez-Rodriguez, I., Basnet, S., Brauner, H., Bobenic, T., Brannen, R., Elrashidy, M., Braaten, M., Amankwah, S. K., and Barr, A.: Using observed soil moisture to constrain the uncertainty of simulated hydrological fluxes, Hydrological Processes, 36, e14465, https://doi.org/10.1002/hyp.14465, 2022.

Razavi, S. and Gupta, H. V.: A multi-method Generalized Global Sensitivity Matrix approach to accounting for the dynamical nature of earth and environmental systems models, Environmental Modelling & Software, 114, 1–11, https://doi.org/10.1016/j.envsoft.2018.12.002, 2019.

---

## Author Comment (AC2)

**Response to Reviewer 2**

**Summary**
This paper explores how to adequately represent wetlands in land surface models, with a case study in White Gull Crrek, Saskatchewan. The authors find that existing parameterizations which ignore groundwater and upland influences to be inadequate and make some recommendations for how to improve the representation of wetlands in LSMs. The authors explore four different model representations, from fully uncoupled to fully coupled. The main conclusion of the paper is that explicit groundwater interactions must be accounted for the adequately represent wetlands in LSMs, with chained and fully coupled modeling approaches working well under different circumstances. Overall I think the paper was interesting, and straightforward to understand. However, I think this paper is only able to make a conceptual argument for this case, rather a quantitative one. The only comparison to observations comes from the comparison of the groundwater level in the upland site, which tests the ability of the model to represent the upland, but not the wetland. Of course, I am sympathetic to the fact that observations of discharge from the fen don't exist, making this a large challenge.

However, I think the paper would be stronger if it could make a quantitative argument for the need for coupled models in representing the wetlands component, rather than a conceptual one. I think some additional analysis looking at either some downstream discharge could be useful for making this point, if such a stream gauge exists. I think correlating such a streamflow with the daily outflow into the river from your model setups could yield an interesting result. Similarly, correlating the $E_f$ from your fen model with latent heat from the FEN tower vs the OJP tower could make a good case for including the coupling between the wetland and upland. Because of these recommendations for additional analysis, I am recommending major revisions.

We are grateful to the reviewer for these thoughtful comments. However, we might not fully agree that our analysis is only conceptual/qualitative and not quantitative. We wish to be upfront about the lack of available field observations from the field site for our study – indeed few, if any, sites have adequate data to fully constrain simulations of lateral subsurface flow. Given this, our intention is to produce a semi-hypothetical model that allows us to quantitatively test alternative model configurations with the objective of determining the circumstances in which simpler models are consistent with more complex ones. It is desirable to make models as simple as possible and no simpler, so our approach provides a systematic basis for understanding what level of simplification is justifiable. We will further elaborate on the data limitations and scope of this work in the revised manuscript to address this important review comment.

We have already compared the simulated ET from the FEN and OJP towers with those produced by our model (Figure 1 and 2). The simulated ET at Fen site (Figure 1) is overestimated in some years, compared with the measured fluxes. This is because of limitations with how MESH simulates evapotranspiration from wetlands (which are not explicitly represented in MESH), and the model always calculates evapotranspiration at the potential evapotranspiration rate. The simulated ET rates at OJP (Figure 2) have a systematic overestimation, with larger errors after 2012. This was explored in Nazarbakhsh et al., 2020, who found that the errors are happening during the melt period – a problem that is yet to be resolved in MESH.

[Figure]

Figure 2: Annual simulated evapotranspiration (ET) rates at wetland-FEN site in blue lines compared with observed ET at FEN flux tower represented in red lines. Years 2011 and 2012 have missing observations.

[Figure]

Figure 2: Annual simulated evapotranspiration (ET) rates at upland-OJP site in blue lines compared with observed ET at OJP flux tower represented in red lines.

**Major comments**

As of section 2, there is no real mention for the digression into coupling LSMs with GWMs as it relates to wetland modeling. Perhaps a few sentences connecting these topics and prior work on this connection would be useful.
Thanks for pointing out this. We will include more literature regarding LSMs and GWMs coupling in the revised manuscript.

I was unclear where the information from the FEN tower actually comes in. At first it seemed like it was the data used to force the wetland model, but later it seemed like OJP was the site where the meteorologic data was taken. Could you clarify which datasets were used to drive which models?

We use the meteorological data from both two towers OJP and the FEN to force our model. The data from OJP flux tower are used to force the upland component (MESH-OJP), while the data from FEN flux tower are used to force the fen component (MESH-FEN). The outputs (such as evaporation and runoff) from the two MESH setups are used to drive our upland-groundwater-fen model. This will be highlighted in the revised manuscript.

Equation 3 shows that there are several parameters, $c_{spill}$, $h_{spill}$, and $n$, whose values don't seem to be given anywhere. I am not sure how to interpret the results in section 5.2 without knowing these values. To me, it looks like the uncoupled model (V1) simply shows no outflow because $h_f$ is always

less than $h_{spill}$, but I am not sure if this is correct. If so, doesn't that correspond to starting the simulation so that $h_{spill}$ is simply set to whatever the initial fen water level is? Wouldn't modifying the parameter values change the results? Even if you do modify these values, of course you will find that you just don't have enough water to maintain the fen's water level because precipitation is not enough to maintain the water level, but showing how these parameters affect this result would be useful and probably highlight your point that the coupling is necessary to maintain predict reailistic discharge from the fen.

Please note – a similar comment was made by reviewer 1 – we provide the same response here. We agree we should provide the parameter values used, and explain how these values were selected, and will do this in the revised manuscript. These three parameters control the timing and magnitude of discharge from the fen into the river channel. We did not have streamflow data to calibrate these parameters, and therefore our approach was to use sensible fixed parameter values that enable us to perform controlled numerical experiments. Our objective is not to simulate streamflow precisely, but rather to explore the sensitive in simulations to changing coupling between the wetland and the upland.

h_spill represents the spilling threshold of the fen, so that when the fen water level is below h_spill, there is not discharge from the fen into the river. We set this value equal to the elevation of the fixed head boundary from the uncoupled groundwater model (Figure 2b/c).

The values of c$_{spill}$ and n (c$_{spill}$ = 0.1 and n = 1.5) were arbitrarily chosen within the reasonable ranges. The ranges were defined based on the recommended ranges in (Razavi and Gupta, 2019), which they used in modelling the fast reservoir with non-linear response in the HBV-SASK model.

We will make this clear in the revised manuscript.

While you say the data is available, the code to reproduce the experiments does not seem to be. This should be linked in the "Data Availability" section.
We will make that available with the revised manuscript.

**Minor comments**

The term $L$ is never defined in equation 2.
L value is the length of the upland hillslope. That will be added.

Line 186-7: "chosen rather arbitrary" -> "chosen arbitrarily"
We will modify this in the revised manuscript.

Line 238: You compare V1 and V3 and don't include V2 because it is functionally the same as V1 for the uplands, but having this spelled out here would here would be useful for anyone who skimmed over 5.1. Maybe just say (V1/V2)?
This is a good suggestion – we will adopt this.

Line 280: "Considering the GW dynamics underneath the upland is essential" Could you elaborate what it is essential for?

This is essential for more accurate simulation of streamflow into river, instead of high overestimation of the streamflow, which happen when ignoring groundwater representation. We will make it more clear in the revised manuscript.

Figure 6: Need to fix "(Error! Reference source not found.)" in the caption.
That should refer to Figure 2 in the manuscript. We will fix this.

**References**

Nazarbakhsh, M., Ireson, A. M., and Barr, A. G.: Controls on evapotranspiration from jack pine forests in the Boreal Plains Ecozone, Hydrological Processes, 34, 927–940, https://doi.org/10.1002/hyp.13674, 2020.

Razavi, S. and Gupta, H. V.: A multi-method Generalized Global Sensitivity Matrix approach to accounting for the dynamical nature of earth and environmental systems models, Environmental Modelling & Software, 114, 1–11, https://doi.org/10.1016/j.envsoft.2018.12.002, 2019.

---

## Author Response (AR1)

Dear Dr. Christa Kelleher,

We sincerely appreciate the time and effort you and the reviewers have dedicated to evaluating our manuscript. Your insights and comments have been invaluable in shaping the quality of our work. We have thoroughly reviewed each of the reviewers' comments and suggestions and we have incorporated all of them into the revised manuscript. Below, we provide a detailed response to each comment, outlining the changes we made and the reasons for our decisions. We look forward to hearing from you.

 Sincerely,

**Response to Reviewer 1**

**General Comments**

Here the authors explore different model complexities and configurations to highlight the need for coupling upland-wetland interactions in land surface models to better capture downstream hydrologic fluxes. In general, the inclusion of wetlands in LSMs is an important consideration for many types of landscapes. Overall, the manuscript was clear and the approach was sound. Below are some points that I think the authors should consider to help clarify some points for the reader.

Dear Reviewer, we would like to thank you for your constructive comments on our manuscript. In this file, we respond to each comment to address your concerns. The response for each point is written in blue.

One broad comment is that there is a bit of a disconnect in how the manuscript was framed (the title, abstract, introduction had a heavier emphasis on wetlands in general) vs the actual analysis and discussion (much larger emphasis on fens, which is the study site). The conclusions were logical based on the data from the fen, but it would be helpful to a) emphasize that fens by definition receive significant amounts of groundwater inputs and b) how other types of wetlands with different levels of GW connectivity could change your conclusions.

We agreed that the framing of the manuscript could be improved. We focus mainly on groundwater dominated fen wetlands, since our case study is a fen, but we also conducted numerical experiments that look at non-groundwater dominated wetlands (section 5.4.2).

- *The definition of fens is introduced on Lines 25 and 28.*
- *The title is modified and in the abstract (Lines 13 and 14), we highlighted that the study mostly focus on groundwater-fed wetlands such as fens. In the introduction (Section 1), The main objective of the study is rephrased to align with the actual analysis introduced in the paper (from Line 76 to 81). Also, in the conclusions, the focus of the study is made clearer on Lines377 and 383.*

**Specific Comments**

- It would be helpful to reference more existing literature on the need for including proper interaction between wetlands and the upland to help strengthen the case for this particular study. While there may not be as many LSM studies that look at this directly, drawing parallels to watershed scale studies, which has quite a few studies in the recent years.

- We certainly agreed and included more literature on watershed scale applications of LSS. *Section 1, 2nd paragraph is modified to include more of recent studies related to our topic (from Line 43 to 59).*

- Minor suggestions for Figure 1
  - Expand part d to be larger to match the other panels for clarity;
  - add the location of POJP piezometer for clearer connection to panel c);
  - if possible, indicate the extent of the fen in part d) just to give some context for the reader. I understand that this can be variable throughout the fen. If the fen goes beyond the transect, ignore this comment!

  The suggested edits are incorporated in Figure 1. For the piezometer location, it was already included in panel c (red dot, POJP). As per the fen width, a you mentioned it is variable throughout the fen, and it was obtained from Google Satellite imagery and therefore, it is difficult to indicate it on the cross section.

- L123: The statement regarding the amount of surface water vs groundwater into the fen seems something unique to this system, and isn't necessarily a feature of the V2 configuration – consider putting this elsewhere. As a side note, because I don't have much context to the wetland:upland ratio or the water balance, the reader might be surprised by this statement. Might be worth indicating the relative areas for the wetland + upland and/or some estimated water balance in the site description

- Agreed. The relative magnitude of the surface water flux is not a feature of only the V2 model configuration, so this text is moved from this part to *Line 122*.
- Regarding the wetland-upland ratio, the total study domain is about 3450 m and the wetland width represent about 150 m of that length. *Lines 94 and 95 are added to clearly describe the transect of the study area*

- L137: does the MESH-CLASS model have saturation-excess and infiltration-excess runoff components to determine runoff vs recharge? Similarly, how does it calculate the R flux? I would not expect a full description of the model, but since this is a major connection to the GW component, it would be helpful to briefly describe it. Also, is snow accumulation and melt modelled in the upland? I assume so, but since it's explicitly mentioned in the fen model, but not here, it can create confusion

The MESH/CLASS model uses an infiltration excess method (based on Green and Ampt) to calculate infiltration. Excess ponded water at the top of soil column is used to calculate overland runoff.

Regarding, snow accumulation and melt at the upland, they are simulated by the MESH model. However, these fluxes are not used directly in our upland-wetland model – rather we use runoff and drainage generated from snowmelt by the MESH/CLASS model (MESH_OJP). For the fen the snowmelt fluxes are used directly in the fen water balance model.

- *Lines 157, 158, and 159 are added to briefly describe the MESH model functionality of infiltration and runoff.*

- L148: I may be mistaken, but I don't think Figure 1 show the groundwater divide/no-flow boundary condition (this is in Fig 2?)

- *This is modified on Line 162.*

- L151 and paragraph: would be helpful to mention how dx is determined within the Darcy's Law calculation

- *That is added on Line 167.*

- L165: Units are not consistent across equation. If Ro is upland runoff (m3/d) and L is the hillslope length (m), the units are [L4/T]; similarly the (Rg+M-Ef)wf component have units of [L2/T], while the Q terms are [L3/T].

Thanks for pointing out this mistake. The units of $RO$ should be m/d and for $Q_G$ and $Qout$ its m²/d, knowing that we assume a unit area (1m) in y direction.

- *This is corrected on Lines 179, 180, and 181.*

- L168: Are Cspill, hspill, n calibrated parameters, and is the model sensitive to them? I believe they are not referenced again later on but one might assume that they have major roles in changing Q.

These three parameters control the timing and magnitude of discharge from the fen into the river channel. We did not have streamflow data to calibrate these parameters, therefore, our approach was to use sensible fixed parameter values that enable us to perform controlled numerical experiments. Our aim is not to precisely simulate streamflow, but rather to investigate how the simulations respond to variations in the coupling between the wetland and the upland.

h_spill represents the spilling threshold of the fen, so that when the fen water level is below h_spill, there is no discharge from the fen into the river. We set this value equal to the elevation of the fixed head boundary from the uncoupled groundwater model (Figure 2b/c).

The values of $c_{spill}$ and n ($c_{spill}$ = 0.1 and n = 1.5) were arbitrarily chosen within the reasonable ranges. The ranges were defined based on the recommended ranges in (Razavi and Gupta, 2019), which they used in modelling the fast reservoir with non-linear response in the HBV-SASK model.

- *That is made clearer in Section 3.4 from Line 191 to 201. To make the main objective more obvious to the readers, Lines 76, 77, and 78 are added in the introduction and Line 14 is added to the abstract.*

- L181: Is there any downstream gage to calibrate? I would be hesitant to say that calibrating the GWT in the upland represents the performance of the collective fen-upland-GW models - especially since three of them do not have the backwards interaction in the model structure

Unfortunately, there is no downstream gauge to calibrate. We agree that calibrating the GWT is not sufficient to conclude that the model accurately represents our specific field site – however, the main

objective in this study is to perform a comparison between the alternative modeling approaches with using a one reasonable parameter set (as we explained in the previous response).

- L185: were these all generated from uniform distributions?

Yes, the 15,000 realizations were generated using a uniform distribution.

- *This is added on Line 208.*

- L186: maybe use "(threshold is chosen arbitrarily based on…)" instead of "chosen rather arbitrary"

- *This is modified on Line 210.*

- L189: I would consider putting in the best parameter set in Table 1 to give context for readers rather than just embedded in the text later on

- *A new column "Calibrated parameters" is added to Table 1.*

- L194: are these L values corresponding to likely wetland-upland areas?

These values correspond to the length of the upland hillslope. At this case we explore the effect of changing the area of upland with having the same area for the fen.

- Figure 3: Should write in caption the simulation number (V1?)

Yes, these results are for model version V1.

- *This is added on Line 247.*

- Figure 6d: missing y axis label to be consistent

- *That is modified in Figure 6.*

- L291: It is slightly hard to understand why at low L values, the groundwater table does not fluctuate in the chained model – it would be worth discussing why. I would assume there's still stochastic inputs to the groundwater from recharge/precipitation, and it's not that the groundwater table has reached the lower boundary/bedrock

In the chained model, when the hillslope length is small, the water table experiences minimal fluctuation due to the presence of a fixed head boundary condition. In contrast, the coupled model lacks a fixed head boundary, leading to more pronounced fluctuations in the water table.

- *The reason for that is added on Line 315.*

- L306: Land is capitalized mid-sentence

- *Modified on Line 329.*

- L306: Because the reader does not know how the forest and grass (is it solely a runoff-coefficient difference, or does ET get affected too?) affects the model fluxes, it's hard to attribute the changes in the model to solely the soil properties, which is the focus of this section.

In our model, changing the vegetation impacts ET and runoff, while changing the soil properties changes the runoff vs infiltration, recharge, and ET. The first model configuration in this numerical experiment was our original setup (OJP in the southern boreal forest). The second configuration (grass over finer soil) was designed to maximize surface runoff, to provide a contrasting case with the groundwater dominated case. The configuration was based on a model configuration for a grassland site (St Denis in Ireson et al., 2022 - MESH point scale paper, in the prairies south of the forest). We did this because we have credible parameters for each configuration – they are not entirely hypothetical.

- *Few lines are added to make this point clearer (From Line 331 to 336).*

- L313: I think that for this instance, it is true that the chained approach is adequate to illustrate the coarse grained soil texture. But I think it's worth commenting that in areas of smaller hillslopes/contributing areas, that may not be the case (as proven in your previous experiment)

Agreed.

- *Line 343 is modified to make this point clearer.*

L346: I would not necessarily include wetlands in 'fen/wetlands' as fens by definition have a lot of GW inputs. Having wetlands here can cause readers to assume that wetlands that either have more bi-directional interactions with the upland via groundwater, or don't receive groundwater, should be treated the same way. While the authors wouldn't run more simulations to capture other wetland types, I think it's a valuable discussion point

Agreed.

- *This point is made clearer in the conclusions on Lines 377 and 383.*

**Response to Reviewer 2**

**Summary**
This paper explores how to adequately represent wetlands in land surface models, with a case study in White Gull Crrek, Saskatchewan. The authors find that existing parameterizations which ignore groundwater and upland influences to be inadequate and make some recommendations for how to improve the representation of wetlands in LSMs. The authors explore four different model representations, from fully uncoupled to fully coupled. The main conclusion of the paper is that explicit groundwater interactions must be accounted for the adequately represent wetlands in LSMs, with chained and fully coupled modeling approaches working well under different circumstances. Overall I think the paper was interesting, and straightforward to understand. However, I think this paper is only able to make a conceptual argument for this case, rather a quantitative one. The only comparison to observations comes from the comparison of the groundwater level in the upland site, which tests the ability of the model to represent the upland, but not the wetland. Of course, I am sympathetic to the fact that observations of discharge from the fen don't exist, making this a large challenge.

However, I think the paper would be stronger if it could make a quantitative argument for the need for coupled models in representing the wetlands component, rather than a conceptual one. I think some additional analysis looking at either some downstream discharge could be useful for making this point, if such a stream gauge exists. I think correlating such a streamflow with the daily outflow into the river from your model setups could yield an interesting result. Similarly, correlating the $E_f$ from your fen model with latent heat from the FEN tower vs the OJP tower could make a good case for including the coupling between the wetland and upland. Because of these recommendations for additional analysis, I am recommending major revisions.

We are grateful to the reviewer for these thoughtful comments. However, we might not fully agree that our analysis is only conceptual/qualitative and not quantitative. We wish to be upfront about the lack of available field observations from the field site for our study – indeed few, if any, sites have adequate data to fully constrain simulations of lateral subsurface flow. Given this, our intention is to produce a semi-hypothetical model that allows us to quantitatively test alternative model configurations with the objective of determining the circumstances in which simpler models are consistent with more complex ones. It is desirable to make models as simple as possible and no simpler, so our approach provides a systematic basis for understanding what level of simplification is justifiable. We elaborated on the data limitations and scope of this work in the revised manuscript to address this important review comment.

We have compared the simulated ET from the FEN and OJP towers with those produced by our model (Figure 1 and 2). The simulated ET at Fen site (Figure 1) is overestimated in some years, compared with the measured fluxes. This is because of limitations with how MESH simulates evapotranspiration from wetlands (which are not explicitly represented in MESH), and the model always calculates evapotranspiration at the potential evapotranspiration rate. The simulated ET rates at OJP (Figure 2) have a systematic overestimation, with larger errors after 2012. This was explored in Nazarbakhsh et al., 2020, who found that the errors are happening during the melt period – a problem that is yet to be resolved in MESH.

[Figure]

Figure 2: Annual simulated evapotranspiration (ET) rates at wetland-FEN site in blue lines compared with observed ET at FEN flux tower represented in red lines. Years 2011 and 2012 have missing observations.

[Figure]

Figure 2: Annual simulated evapotranspiration (ET) rates at upland-OJP site in blue lines compared with observed ET at OJP flux tower represented in red lines.

**Major comments**

As of section 2, there is no real mention for the digression into coupling LSMs with GWMs as it relates to wetland modeling. Perhaps a few sentences connecting these topics and prior work on this connection would be useful.

- *Section 1, 2ⁿᵈ paragraph is modified to include more of recent studies related to our topic.*

I was unclear where the information from the FEN tower actually comes in. At first it seemed like it was the data used to force the wetland model, but later it seemed like OJP was the site where the meteorologic data was taken. Could you clarify which datasets were used to drive which models?

We use meteorological data from two towers, OJP and FEN, to drive our model. The data from the OJP flux tower is employed to force the upland component (MESH-OJP), while the data from the FEN flux tower is utilized to force the fen component (MESH-FEN). The outputs, including evaporation and runoff, from both MESH setups are employed as inputs to drive our upland-groundwater-fen model.

- *This is made clearer in Section 3.2 on Lines 150 and 151 and in Section 3.4 on Lines 182 and 183, by adding the names of each MESH model setup.*

Equation 3 shows that there are several parameters, $c_{spill}$, $h_{spill}$, and $n$, whose values don't seem to be given anywhere. I am not sure how to interpret the results in section 5.2 without knowing these values. To me, it looks like the uncoupled model (V1) simply shows no outflow because $h_f$ is always less than $h_{spill}$, but I am not sure if this is correct. If so, doesn't that correspond to starting the simulation so that $h_{spill}$ is simply set to whatever the initial fen water level is? Wouldn't modifying the parameter values change the results? Even if you do modify these values, of course you will find that you just don't have enough water to maintain the fen's water level because precipitation is not enough to maintain the water level, but showing how these parameters affect this result would be useful and probably highlight your point that the coupling is necessary to maintain predict reailistic discharge from the fen.

Please note – a similar comment was made by reviewer 1 – we provide the same response here. These three parameters control the timing and magnitude of discharge from the fen into the river channel. We did not have streamflow data to calibrate these parameters, and therefore our approach was to use sensible fixed parameter values that enable us to perform controlled numerical experiments. Our aim is not to precisely simulate streamflow, but rather to investigate how the simulations respond to variations in the coupling between the wetland and the upland.

h_spill represents the spilling threshold of the fen, so that when the fen water level is below h_spill, there is not discharge from the fen into the river. We set this value equal to the elevation of the fixed head boundary from the uncoupled groundwater model (Figure 2b/c).

The values of $c_{spill}$ and n ($c_{spill}$ = 0.1 and n = 1.5) were arbitrarily chosen within the reasonable ranges. The ranges were defined based on the recommended ranges in (Razavi and Gupta, 2019), which they used in modelling the fast reservoir with non-linear response in the HBV-SASK model.

- *That is made clearer in Section 3.4 from Line 193 to 203. To make the main objective clearer to the readers, Lines 76, 77, and 78 are added in the introduction and Line 14 is added to the abstract.*

While you say the data is available, the code to reproduce the experiments does not seem to be. This should be linked in the "Data Availability" section.

- *A link to a repository of the study's scripts is added in the "Data Availability" section.*

**Minor comments**

The term $L$ is never defined in equation 2.
L value is the length of the upland hillslope.

- *A definition of "L" is added on Line 179.*

Line 186-7: "chosen rather arbitrary" -> "chosen arbitrarily"

- *This is modified on Line 210.*

Line 238: You compare V1 and V3 and don't include V2 because it is functionally the same as V1 for the uplands, but having this spelled out here would here would be useful for anyone who skimmed over 5.1. Maybe just say (V1/V2)?
Agreed.

- *That is added on Lines 261 and 262.*

Line 280: "Considering the GW dynamics underneath the upland is essential" Could you elaborate what it is essential for?
This is essential for more accurate simulation of streamflow into river, instead of high overestimation of the streamflow, which happen when ignoring groundwater representation.

- *That is made clearer on Line 303.*

Figure 6: Need to fix "(Error! Reference source not found.)" in the caption.
That should refer to Figure 2 in the manuscript.

- *This is fixed on Line 306.*

**References**

Ireson, A. M., Sanchez-Rodriguez, I., Basnet, S., Brauner, H., Bobenic, T., Brannen, R., Elrashidy, M., Braaten, M., Amankwah, S. K., and Barr, A.: Using observed soil moisture to constrain the uncertainty of simulated hydrological fluxes, Hydrological Processes, 36, e14465, https://doi.org/10.1002/hyp.14465, 2022.

Nazarbakhsh, M., Ireson, A. M., and Barr, A. G.: Controls on evapotranspiration from jack pine forests in the Boreal Plains Ecozone, Hydrological Processes, 34, 927–940, https://doi.org/10.1002/hyp.13674, 2020.

Razavi, S. and Gupta, H. V.: A multi-method Generalized Global Sensitivity Matrix approach to accounting for the dynamical nature of earth and environmental systems models, Environmental Modelling & Software, 114, 1–11, https://doi.org/10.1016/j.envsoft.2018.12.002, 2019.

---

## Author Response (AR2)

Dear Dr. Christa Kelleher,

We sincerely appreciate the time and effort you and the reviewers have dedicated to evaluating our manuscript. Below is our response to the minor change requested by Reviewer 1.

 Sincerely,

**Response to Reviewer 1**

Overall, the authors addressed the reviewers' concerns sufficiently and I believe that the manuscript is much stronger and clearer as a result.

While I agree that having a downstream gage would add to the paper (I also mentioned that in my initial comments), I don't think that the work here should necessarily be discounted. I feel that the revised manuscript provides sufficient framing of their (limited) available data and objectives to justify the scope of their work and approach. I would encourage the authors to put the ET comparisons seen in the Author Response in the supplemental materials, and refer to them in the manuscript as they would be helpful in supporting the validity of the model.

A supplemental file is uploaded which includes the ET results at OJP and FEN.

- *The figures are refrenced in the paper on Line 258.*